# Application of Pharmacometrics in Pharmacotherapy: Open-Source Software for Vancomycin Therapeutic Drug Management

**DOI:** 10.3390/pharmaceutics11050224

**Published:** 2019-05-09

**Authors:** Soo Hyeon Bae, Dong-Seok Yim, Hyemi Lee, Ae-Ryoung Park, Ji-Eun Kwon, Hirata Sumiko, Seunghoon Han

**Affiliations:** 1PIPET (Pharmacometrics Institute for Practical Education and Training), College of Medicine, The Catholic University of Korea, 222 Banpodaero Seocho-gu, Seoul 06591, Korea; shbae@kirams.re.kr (S.H.B.); yimds@catholic.ac.kr (D.-S.Y.); hyemi.lee@catholic.ac.kr (H.L.); 2Department of Pharmacology, College of Medicine, The Catholic University of Korea, 222 Banpodaero, Seocho-gu, Seoul 06591, Korea; 3Department of Pharmacy, Seoul St. Mary’s Hospital, The Catholic University of Korea, 222 Banpodaero Seocho-gu, Seoul 06591, Korea; parkar@cmcnu.or.kr (A.-R.P.); csoko83@cmcnu.or.kr (J.-E.K.); hiratasumiko@hanmail.net (H.S.)

**Keywords:** Korean, vancomycin, therapeutic drug management, population pharmacokinetics, open-source software

## Abstract

The population pharmacokinetic (PK) parameters that are implemented in therapeutic drug management (TDM) software were generally obtained from a Western population and might not be adequate for PK prediction with a Korean population. This study aimed to develop a population PK model for vancomycin using Korean data to improve the quality of TDM for Korean patients. A total of 220 patients (1020 observations) who received vancomycin TDM services were included in the dataset. A population PK analysis was performed using non-linear mixed effects modeling, and a covariate evaluation was conducted. A two-compartment model with first-order elimination best explained the vancomycin PK, with estimates of 2.82 L/h, 31.8 L, 11.7 L/h, and 75.4 L for *CL*, *V_1_*, *Q,* and *V_2_*, respectively. In the covariate analysis, weight correlated with the volume of the peripheral compartment, and creatinine clearance, hemodialysis, and continuous renal replacement therapy treatments contributed to the clearance of vancomycin. The results show the clear need to optimize the PK parameters used for TDM in Korean patients. Specifically, *V_1_* should be smaller for Korean patients, and renal replacement therapies should be considered in TDM practice. This final model was successfully applied in R shiny as open-source software for Koreans.

## 1. Introduction

Vancomycin (VCM) is widely used for the treatment of infectious diseases including bacteremia, endocarditis, pneumonia, and meningitis, and it is the first-line agent for the treatment of methicillin-resistant strains of *Staphylococcus* [1,2,3]. After intravenous infusion, VCM is rarely metabolized and is primarily excreted unchanged in urine by glomerular filtration. Thus, renal function is one of the most important factors influencing patient exposure to VCM [4]. The elimination half-life with normal renal function is reported to be from 3 to 9 h [5].

When patients receive VCM treatment, therapeutic drug management (TDM) is generally recommended because VCM exhibits clear exposure–response relationships and has exposure-related nephrotoxicity [6]. Although the ranges of desired concentration differ depending on the strains and site of infections, the general target trough concentrations are 10 to 20 mg/L [2,7]. Several papers have reported that trough concentrations greater than 15 mg/L [8] or 12 mg/L [9] increase the risk of nephrotoxicity. Therefore, dose adjustment is needed depending on the patient’s renal function and concomitant nephrotoxic medications. In other studies, the area under the plasma concentration–time curve up to 24 h (*AUC_24_*)/minimum inhibitory concentration (MIC) or *AUC_24_* ≥ 400 mg h/L was suggested as the target exposure for VCM to optimize the response to therapy [6,10,11]. 

In clinical settings, Bayesian feedback TDM is widely applied, and peak plasma concentration (*C_peak_*) and trough plasma concentration (*C_trough_*) are usually monitored for target attainment. However, it has been reported that the concentrations of VCM predicted by commercially available software tended to be higher than the observed concentrations in patients with low renal function, low body weight, or old age [12]. The population pharmacokinetic (PK) parameters implemented in TDM software were generally obtained in Western populations and might be inappropriate for PK prediction in a Korean population (who has different demographic characteristics in comparison to the Western population). Moreover, patients who received hemodialysis (HD) or continuous renal replacement therapy (CRRT) had significantly higher VCM exposures, and to our knowledge, no software that is currently available can consider the patient’s HD or CRRT treatment status. 

Our aims in this study were (1) to develop a population PK model for VCM using Korean data, and (2) to establish open-source TDM software for Korean patients using our final PK model. In this study, we have accounted for renal function in the clearance (*CL*) parameters and incorporated changes to *CL* according to HD and CRRT treatment as well as the patient’s estimated creatinine clearance (CL_CR_).

## 2. Materials and Methods 

### 2.1. Ethics

This study protocol was approved by the Institutional Review Board of Seoul St. Mary’s Hospital (KC19RESK0107, date of approval: 17 FEB 2018). The investigators performed the study in accordance with all applicable ethical standards and local regulations.

### 2.2. Dataset

VCM plasma concentration data from patients who received VCM TDM services from June 2015 to August 2017 in Seoul St. Mary’s Hospital were included in the dataset for this population PK analysis. Demographic and clinical variables of the patients—age, sex, height, body weight (WT), albumin, urea nitrogen, infection type, CL_CR_ calculated by the Cockcroft–Gault equation, and the application of HD and CRRT—were collected for evaluation as potential covariates [13]. 

### 2.3. Population PK Modeling

A population PK analysis was performed using the first-order conditional estimation method with interaction in non-linear mixed effects modeling (NONMEM, version 7.4, ICON Development Solution, Ellicott City, MD, USA) and RStudio (version 0.99). R (version 3.2.2) and Xpose4 (version 4.5.3) were used for all graphical analyses and model diagnostics. IV infusion with one- and two-compartment models was tested, and previously reported population PK models were also considered in constructing this model [5]. The results of the likelihood ratio testing were deemed significant if decreases in the objective function value were more than 3.84 (*p* < 0.05, *df* = 1) or 5.99 (*p* < 0.05, *df* = 2). The appropriateness of the model was evaluated using goodness-of-fit plots, condition number, eta (inter-individual random variable) shrinkage, successful convergence, matrix singularity, and likelihood ratio testing. VCM concentrations of more than 100 ng/mL were considered outliers and excluded from modeling. The between-subject variability of each parameter was described using a log-normal distribution as: P_i_ = θ · exp(η_i_)(1) where P_i_ represents the parameter (e.g., *CL* or *V*) for an individual, θ is the typical value of the model parameter, and η_i_ is the inter-individual random variable, assumed to have a normal distribution with a mean of zero and variance of ω^2^. The covariance between the random effects was tested using the OMEGA BLOCK option. The combined error model was applied to explain residual error: (2)$$Y_{\mathrm{ij}}=\mathrm{IPRE}D_{\mathrm{ij}}+\sqrt{\sigma_{1}^{2}+\sigma_{2}^{2}\cdot \mathrm{IPRE}D_{\mathrm{ij}}^{2}}\;\times \varepsilon_{\mathrm{ij}}$$ where Y_ij_ denotes the observed concentration of the ith individual at time j, IPRED_ij_ is the corresponding predicted concentration for Y_ij_, ε_ij_ is the intra-individual random variable distributed with a mean of zero and variance of *σ*^2^, and *σ*_1_^2^ and *σ*_2_^2^ represent additive and proportional variance components, respectively. The coefficient of variation (CV) is written as:(3)$$\mathrm{CV}(\%)\;=\;\sqrt{\exp(\omega^{2})-1}\;\times 100$$

### 2.4. Covariate Evaluation

The demographic and clinical variables of the patients listed in Table 1 were evaluated for their influence on the VCM PK parameters. For covariate screening, both visual (parameter vs. covariate scatterplots for continuous variables and boxplots for discrete variables) and numerical (generalized additive model in Xpose, version 4.5.3) screening procedures were performed. Covariates were then selected using forward selection–backward elimination with a likelihood ratio test (forward selection *p* < 0.05 and backward elimination *p* < 0.01).

### 2.5. Model Evaluation and Simulation

Since the PK model developed in this study was to be used as a backbone of Bayesian feedback TDM, the evaluation of predictive performance was essential. However, because each patient included in the dataset had a different dosage regimen, observation time, and covariate values, a traditional visual predictive check (VPC) was not suitable. Therefore, a prediction- and variability-corrected VPC (pvcVPC) suggested by Bergstrand et al. was conducted [13]. In this procedure, a bin was defined according to the independent variables (time, dose, and other significant covariate values), and the observed values were corrected using the following Equation (4): (4)$$\ln\left(\mathrm{pvc}Y_{ij}\right)=\;\ln\left(Y_{ij}\right)+(\ln\left(\tilde{PRED_{bin})}-\ln\left(PRED_{ij}\right)\right)$$ where $\mathrm{pvc}Y_{ij}$ is prediction- and variability-corrected observation of prediction for the *i*th individual at the *j*th time point, $\tilde{PRED_{bin}}$ is the median of typical population predictions for the specific bin of independent variables, and $PRED_{ij}$ represents typical population predictions for the *i*th individual at the *j*th time point. Perl-speaks-NONMEM (PsN, version 4.8.1, downloaded at https://uupharmacometrics.github.io/PsN/install.html), Pirana (version 2.9.9, Certara), and RStudio (version 0.99) with R (version 3.2.2) were used for the pvcVPC. A total of 1000 replicates of the simulated dataset were produced and used for the determination of prediction intervals. 

To evaluate the robustness of the parameters obtained from the final model, we performed 1000 non-parametric bootstrap replications with the final PK model using Wings for NONMEM (version 741). The median and 95% confidence intervals of parameters from the bootstrap resampling were compared with the final parameter estimates. The steady-state PK profiles of VCM under several different conditions, including covariates, were then simulated. A total of 1000 virtual subjects in each group were simulated after a daily 1 g administration of VCM for 7 days (1-h IV continuous infusion), and the median and 90% prediction intervals were reported.

### 2.6. R Shiny Application for VCM TDM

After we completed the final population PK model for VCM, we used the open-source R Shiny program to establish VCM TDM software. The structural PK model and the final estimates of both fixed-effect and random-effect parameters (in the form of a matrix of *θ*, *ω*^2^, and *σ*^2^) were translated into an R code. For the estimation of empirical Bayes estimates (EBEs) for the set of PK parameters (*η*s, the difference of the individual patient from the value of the population PK parameter) for the patient of interest, the information on basic patient characteristics (which was implemented as covariates for the PK model), dosing history, and plasma concentration values with sampling time records were required as the user input values. The optimal estimates for EBEs were obtained by minimizing the value of the objective function defined as Equation (5) [14,15]: (5)$$O(\vec{\eta_{i}})\;=\;-2LL\left(\vec{\eta_{i}}\right)\;=\sum_{j}\left[log\sigma_{ij}^{2}+\frac{{\left(Y_{ij}-\;F_{ij}\right)}^{2}}{\sigma_{ij}^{2}}\right]+\;\vec{\eta_{i}}{\;}^{T}\Omega^{-1}\;\vec{\eta_{i}}$$ where $O\left(\vec{\eta_{i}}\right)$ and $LL\left(\vec{\eta_{i}}\right)$ represent the objective function and log-likelihood values dependent on the set of EBEs $\left(\vec{\eta_{i}}\right)$ in the corresponding patient, respectively. Since $Y_{ij}$ (observed concentrations) and $F_{ij}$ (predicted concentrations) are determined by the user input values and $\sigma_{ij}^{2}$ and $\Omega^{-1}$ (derived from ω^2^ matrix) are given in the PK model, users may obtain optimized values of PK parameters for a patient with this algorithm. In addition, with the obtained EBE values, we could implement a graphical presentation for predicted concentration over time under a planned dosage regimen as well as under the current regimen for users to select the best dosage modification strategy. To convert a vector into a full matrix for computing the log-likelihood function in R, the codes in the article by Kim et al. [15] were applied in the script. Several R packages (deSolve, trustOptim, DEoptim, plyr, dplyr) were installed and utilized to build R codes for the algorithm explained above.

## 3. Results

### 3.1. Population PK Modeling

A total of 1020 VCM observations from 220 patients that received VCM TDM services were included in the dataset. Patient demographic information is summarized in Table 1. A two-compartment, first-order elimination model best described the PK of VCM. In the results from the covariate analysis, CL_CR_ was included as a statistically significant covariate of CL, and WT was selected as a covariate of V_2_. The CLs of patients who received CRRT or HD treatment at the time of VCM therapy were 0.716 and 0.334, respectively, indicating that they differed significantly from those of patients not receiving CRRT or HD (with a CL_CR_ of 72 ml/min, the estimated CL was 2.80 L/h). The typical values in volume terms were estimated as 31.8 for V_1_ and 75.4 for V_2_ (assuming WT = 60 kg). The goodness-of-fit plots of the PK model with all covariates suggest that the final model adequately describes the observed data (Figure 1). The final estimated PK parameters are summarized in Table 2.

### 3.2. Model Assessment and Simulation

Automated binning by PsN was firstly attempted to conduct pvcVPC. However, considering the duration of VCM TDM services in practice and that the majority of samples focused on the early period of treatment, the final pvcVPC was conducted with data binned at 0, 8.5, 12.5, 24.5, 48.5, 76, 100, 200, 300, 400, 500, 600, 700, 800, and 900 h time intervals. The result of pvcVPC is shown in Figure 2, which was generated using a modified R code originally provided at PMX Solutions website [15]. The predictive performance of the final PK model was considered to be acceptable. This was based on the pvcVPC result that most of the observations were within the 90% prediction interval regardless of covariate values, particularly in the early period (<250 h) after the first dose. An inflation in the width of prediction interval (with the inflated confidence intervals for the margin of prediction interval) was observed due to an insufficient number of observations included in the original dataset. For example, there were only 19 observations from five patients in the period later than 600 h after the first dose.

The median parameter estimates from the 1000 bootstrap replication data were similar to those from the final model, whose results are summarized in Table 2. Because those results suggested that the predictive performance of the final PK model was appropriate, we performed various simulations and applications based on different conditions of statistically meaningful covariates such as CL_CR_, HD, and CRRT, and all patients were assumed to weigh 60 kg. 

Dosage regimen: daily 1 g VCM for 7 days, 1-h IV continuous infusion 

Group 1: patients with 100 mL/min CLCR (normal renal function) 

Group 2: patients with 40 mL/min CLCR (moderate renal impairment) 

Group 3: patients receiving CRRT treatment during VCM therapy 

Group 4: patients receiving HD treatment during VCM therapy 

In addition to the pvcVPC, the steady-state PK profiles for VCM were simulated, and the median and 90% prediction intervals are shown in Figure 3. Representative targets for *C_peak,ss_* and *C_trough,ss_* (50 mg/L and 10 mg/L, respectively) are shown in the same graph. The simulated *C_peak,ss_* and *C_trough,ss_* were 34 mg/L and 8 mg/L, respectively, in Group 1, and 14 mg/L and 40 mg/L, respectively, in Group 2. In the CRRT and HD treatment groups, the overall exposure to VCM was higher than in patients not receiving CRRT or HD. We also performed additional simulations with different weight values and confirmed that the results were acceptable regardless of the weight ranges. (Data are not shown).

### 3.3. R Shiny Application for VCM TDM

The two-compartment structure model for VCM, the final estimates of both fixed effects and random effects from the PK model (Table 2), and the log-likelihood Equation (5) for estimating *η_i_* were implemented in R script. A draft version of the software is now available at http://pipet.shinyapps.io/vancomycin. Figure 4 shows examples of the R Shiny app VCM TDM interface to simulate how PK profiles can be used to adjust dosage regimens for VCM.

## 4. Discussion

The purposes of this study were (1) to build the population PK model of VCM for Korean patients, and (2) to develop open-source TDM software for VCM. It is well known that not only the toxicity but also the efficacy of VCM is related to the exposure of the drug, and thus the need for TDM of VCM has been emphasized through numerous articles. Nevertheless, for the following reasons, this study is different from the previous research. Firstly, this PK model was constructed based on a large amount of data from patients with severe and moderate renal function. In the patient data used to construct the model, the median and the 25th percentile (Q1) of CL_CR_ calculated by the Cockcroft–Gault formula were 72.4 mL/min and 40 mL/min, respectively. With covariate analysis, body weight for *V_2_*, and CL_CR_ for *CL* were significant and these findings were consistent with the results from previous studies [2,4,7,9]. In addition, HD and CRRT were included as significant covariates of *CL* in this model, allowing more accurate *CL*s for patients with renal dysfunction. The final estimates of *CL* in each group were 2.82 L/h for normal renal function, whereas those for HD therapy and CRRT therapy were 0.716 L/h and 0.334 L/h, respectively. The assessment of the predictive performance of the final PK model was shown as the basic goodness-of-fit plots in Figure 2. For validation of the model, the median parameter estimates obtained from the 1000 bootstrap data were very similar to the estimates from the final PK model. 

With the final PK model, simulation studies reflecting different renal functions and different body weights were conducted. All groups received the same dosage regimen—daily 1 g VCM for 7 days—and according to the results, for patients with CL_CR_ > 100 mL/min (i.e., patients with normal renal function) dose adjustments, such as increasing the dose or dosing intervals, would be required to achieve target concentrations that could be changed by infection type and site. On the other hand, dose reduction of the patients who received HD or CRRT should be carefully considered because the increased concentrations of VCM can induce concentration-related toxicity such as nephrotoxicity. 

Based on these results, TDM software for Koreans was constructed with R Shiny, an open-source program. For TDM, it is essential to predict individual PK profiles by reflecting individual concentration, and Bayesian maximum a posteriori (MAP) estimation is widely used for this. In NONMEM, the objective function (Equation (5)) was used for estimating the post hoc *η* of patients, and this is easily obtained using the POSTHOC option [14]. In this study, to reproduce the estimation method for post hoc *η*, the objective function was incorporated into the code, and some parts of the code were taken from Kim’s study [16] to convert a vector notation to a full matrix. In the R script, the “optim” function was used to optimize the post hoc *η*s of *CL* and *V_1_* for a patient. In the R shiny app, not only patient information on CL_CR_ (which can be directly calculated in the app if the age, sex, weight, and serum creatinine level of the patient are available) and weight, but also observation data of VCM should be entered for PK simulation. Based on the simulation results, an adaptive dosage regimen including infusion duration time, dosing amount, and dosing interval could be applied to simulate further PK profiles. In addition to dosage regimen, the target therapeutic range may be adjusted by the user according to the indication. The R shiny app for VCM TDM can be easily accessed online without a local installation.

Most of the commercially available TDM software for VCM have incorporated population PK parameters obtained from Caucasian patients, and no software based on Korean data can be found so far. The development of TDM software using Korean PK parameters has a great significance itself and furthermore, this interface and the code used for VCM TDM could be applied to many other drugs that require concentration monitoring. This is web-based software which is easy to access, and anyone can use it anywhere, anytime, free of charge. However, this study has several limitations to consider. Because the study was not a strictly controlled clinical trial but a retrospective investigation with data produced in clinical settings, the explained (wide ranges of weight, and renal functions) and unexplained variability (expressed as the value of *ω_CL_)* between subjects was relatively large. Even though the influences of some patient factors were well reflected in the PK model, to avoid an abnormally large or small EBE estimated by the system, the plasma concentration values and the actual sampling time data should be accurately given. In addition, due to the insufficiency of the source data, the credibility of the system in the extended dosing period (e.g. later than 20 days after the initiation of treatment) is not clear. In order to improve the robustness and accuracy of the model, further study should be performed to acquire more data especially on patients with CRRT and HD therapy.

## Figures and Tables

**Figure 1 pharmaceutics-11-00224-f001:**
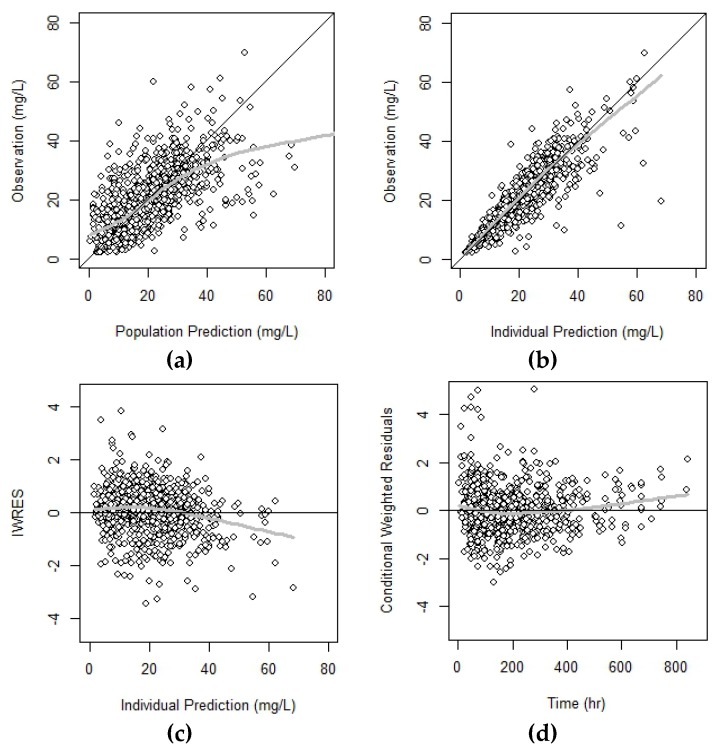
The basic goodness-of-fit plot for the vancomycin pharmacokinetic model. (**a**) Observations versus population predictions, (**b**) observations versus individual predictions, (**c**) individually weighted residuals versus individual predictions, and (**d**) conditional weighted residuals versus time. The solid black y = x or y = 0 lines are the line of identity and the line of reference, respectively. The solid gray lines are the lines of locally weighted scatterplot smoothing (LOWESS), and IWRES refers to the absolute value of individual weighted residuals.

**Figure 2 pharmaceutics-11-00224-f002:**
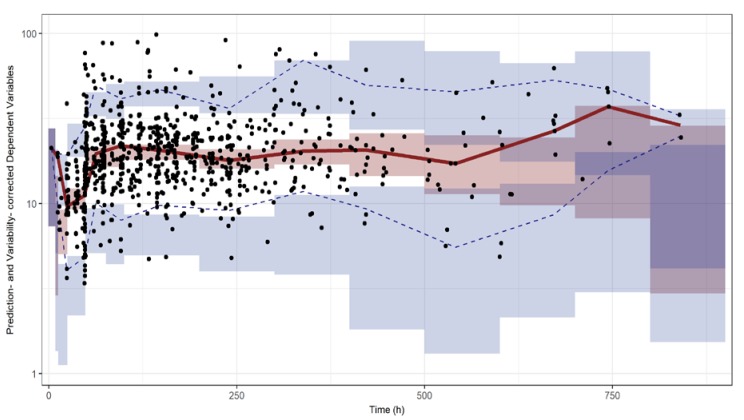
Prediction- and variability-corrected VPC plot. The dots are corrected observations, the solid red line represents the median of corrected observations, the dashed blue lines are 5% and 95% percentiles of corrected observations, the red field represents 95% confidence intervals for the prediction median, and blue fields represent the 95% confidence intervals for the margin of the 90% prediction interval.

**Figure 3 pharmaceutics-11-00224-f003:**
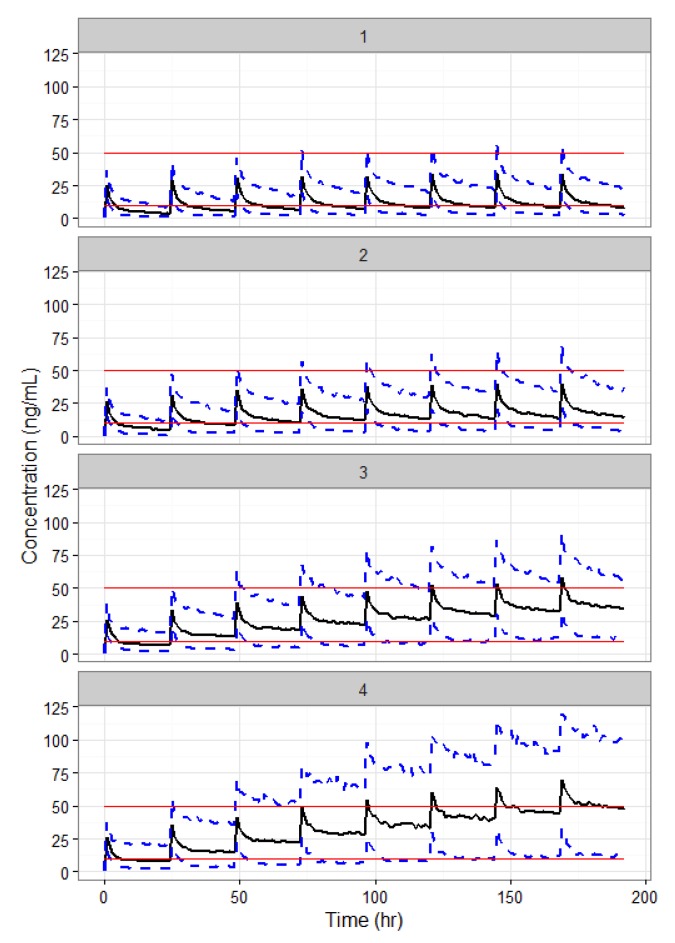
Simulations for plasma vancomycin concentration by virtual patient groups receiving daily 1 g vancomycin (1-h intravenous infusion). Group 1: patients with estimated serum creatinine clearance of 100 mL/min, Group 2: patients with estimated serum creatinine clearance of 40 mL/min CL_CR,_ Group 3: patients receiving continuous renal replacement during vancomycin therapy, Group 4: patients receiving hemodialysis during vancomycin therapy. The black solid line represents the median, the blue lines are 90% prediction intervals, and the red lines are the representative target *C_peak_* (50 mg/L) and *C_trough_* (10 mg/L) of VCM.

**Figure 4 pharmaceutics-11-00224-f004:**
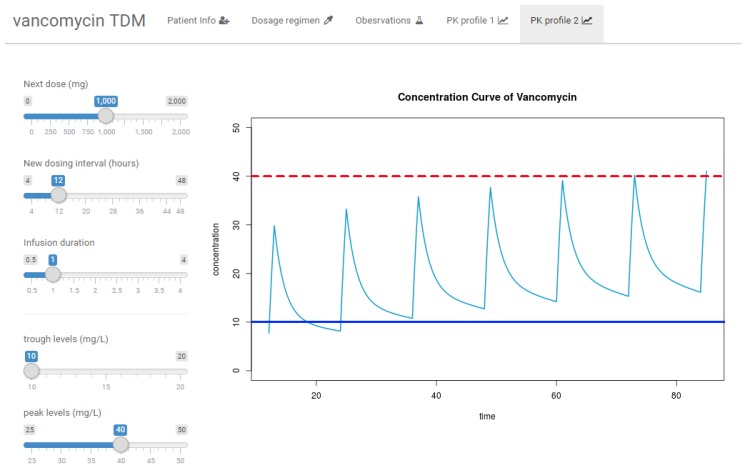
User interface of the open-source vancomycin therapeutic drug management software.

**Table 1 pharmaceutics-11-00224-t001:** Demographic and clinical characteristics of subjects.

Variables (Unit)	Mean (Range)
Age (year)	63 (21–98)
Sex (male/female)	139/81
Weight (kg)	61.6 (30.0–126.7)
Serum creatinine, Scr (mg/dL)	1.7 (0.20–13.3)
Creatinine clearance, CL_CR_ (mL/min) ^1^	77.0 (4.57–279)
Application of continuous renal replacement therapy (CRRT)	9
Patients who received hemodialysis (HD)	20

^1^ Calculated by Cockcroft–Gault equation.

**Table 2 pharmaceutics-11-00224-t002:** Final estimates of population pharmacokinetic parameters.

Parameter	Description	Estimate	%RSE	Bootstrap Median (95% CI)
Structural model				
$CL=\;\theta_{1}\cdot {\left(\frac{CL_{CR}}{72}\right)}^{\theta_{2}}\;$(L/h)	Clearance in patients not receiving CRRT nor HD treatment			
*θ_1_*		2.82	4.18	2.80 (2.56–3.04)
*θ_2_*		0.836	6.89	0.837 (0.717–0.971)
*CL_CRRT_* (L/h)	*CL* in patients with CRRT	0.716	11.0	0.733 (0.437–1.72)
*CL_HD_* (L/h)	*CL* in patients with HD	0.334	11.9	0.335 (0.142–0.452)
*V_1_* (L)	Volume of central compartment	31.8	7.01	32.8 (25.6–42.8)
*Q* (L/h)	Intercompartmental clearance	11.7	7.42	11.3 (6.93–13.8)
$V_{2}=\;\theta_{3\;}\cdot (\frac{Weight}{60}$) (L)	Volume of peripheral compartment			
*θ_3_*		75.4	7.91	75.7 (58.6–94.9)

Inter-individual variability				
*ω_CL_* (%)	Interindividual variability of *CL*	99.2	6.55	101 (83.4–116)
*ω_V2_* (%)	Interindividual variability of *V_2_*	49.2	3.08	48.8 (40.5–57.4)

Residual error				
*σ_prop_*	Proportional error	0.253	2.91	0.250 (0.222–0.281)

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
