# Peer review of "Application of Pharmacometrics in Pharmacotherapy: Open-Source Software for Vancomycin Therapeutic Drug Management"

_pharmaceutics, 2019, doi:10.3390/pharmaceutics11050224_

Round 1
Reviewer 1 Report
The authors developed a population PK model for vancomycin in Korean population, and implemented the analysis results in a R shiny app for therapeutic drug management. The analysis is well performed and the further application of the results is interesting and promising. Some minor/moderate suggestions are listed below for the authors to consider.
1. (Moderate) Before considering using the model for simulation, the adequacy of the final model in the sense of predicting needs to be assessed. So please provide some simulation-based diagnostic plots (visual predictive check or posterior predictive check), which will 1) assess the fitting of the overall trend 2) assess the estimation of random effects, especially inter-individual variability.
2. (Moderate) As a follow-up question, the inter-individual variability (IIV) from the current model is relatively big (~50% and 100%), which also reflects in the huge PI in Figure 2. With such wide variation/uncertainty, it will be very difficult to predict individual profile. So I'm wondering whether the authors have any thoughts on utilizing the model with high IIV for TDM. Please cover this issue in the Discussion section.
3. (Minor) Section 2.3 is missing
4. (Minor) Please add the software and version in model description part.
5. (Minor) Section 3.2 As body weight is also included as a covariate, please explain why not including scenarios with different weight groups for simulation.
Author Response
Thank you for the precious comments.
Reviewer 1
1. (Moderate) Before considering using the model for simulation, the adequacy of the final model in the sense of predicting needs to be assessed. So please provide some simulation-based diagnostic plots (visual predictive check or posterior predictive check), which will 1) assess the fitting of the overall trend 2) assess the estimation of random effects, especially inter-individual variability.
Thank you very much for your valuable comment. We fully agree with your opinion on the necessity of simulation-based diagnostics. Unfortunately, because the data for modeling includes various dosing regimens in each patient with different weight and kidney function, a general VPC with all dosing regimens containing all ranges of covariates was not possible. We, therefore, utilized the method of ‘pvcVPC’. The results were added in Methods (line 111~127) and Results (line 186~203) sections and a related reference (No. 13, Bergstrand M, Hooker AC, Wallin JE, and Karlsson MO. Prediction-corrected visual predictive checks for diagnosing nonlinear mixed-effects models. The AAPS J. (2011) 13:143-151.) was also added. In addition to VPC, 1,000 replication bootstrapping results were provided in Table 2 which was conducted as model assessment.
2. (Moderate) As a follow-up question, the inter-individual variability (IIV) from the current model is relatively big (~50% and 100%), which also reflects in the huge PI in Figure 2. With such wide variation/uncertainty, it will be very difficult to predict individual profile. So I'm wondering whether the authors have any thoughts on utilizing the model with high IIV for TDM. Please cover this issue in the Discussion section.
Thank you for your comments. The inter-individual variability of CL is relatively high in this model and we considered this issue as our limitations of this study. We added this issue in the Discussion (line 285~293).
3. (Minor) Section 2.3 is missing
We are terribly sorry for the error in manuscript editing. The deleted part of Section 2.2. and contents for Section 2.3 were added in the revised manuscript (line 72~91).
4. (Minor) Please add the software and version in model description part.
The details for used software and its version were originally the contents of Section 2.3. Now, it is fully specified in the manuscript (line 77~80 and line 124~126). Thank you.
5. (Minor) Section 3.2 As body weight is also included as a covariate, please explain why not including scenarios with different weight groups for simulation.
Thank you for your comments. Body weight is a covariate of V2 and did not significantly affect the validity of PK prediction performance (i.e. 60 kg vs. 100 kg). We added this in the Results section (line 219~221).
Reviewer 2 Report
The authors of the manuscript titled “Application of Pharmacometrics in Pharmacotherapy: Open-source Software for Vancomycin Therapeutic Drug Management” present a study where a population PK model of vancomycin in a Korean population was developed and a Bayesian dosing tool was developed in R Shiny.
This paper is very interesting and potentially very useful and of interest to the Pharmaceutics readership. However, issues with methodology may potentially affect the study’s validity.
I have a few questions and comments for the authors to consider.
Main comments:
1. The manuscript suffers from several editorial issues. I have described minor issues under Minor comments (below) but there appears to be paragraphs missing which makes an assessment of the scientific merit of this manuscript difficult (e.g. in the Methods section). I suggest enlisting the help of a proof-reader.
2. One of the most useful model evaluation techniques is a VPC plot. The GOF plots can be visually deceptive of model performance particularly with repeat data such as yours. I suggest you create a VPC plot
3. It is not clear, perhaps missing?, how R Shiny was used was a dosing tool once the PK model was developed and coded in R Shiny. It seems you obtained the EBEs from NONMEM then added them to R Shiny? If so, this isn’t sensible as the EBE pertain to the individual so you can only perform TDM on that individual only!?
4. Also not clear whether a Bayesian estimator was added to R Shiny and the nature of the estimator (was it MAP?).
5. Also related to above, which ODE solver was used? I think general technical information about the dosing tool are missing in this manuscript.
Minor comments:
6. There were a few typographical, grammar, and sentence structure issues throughout the manuscript. A couple of examples below. I recommend enlisting the help of a proof-reader.
a. Page 1, Line 36: “osteomyelitis meningitis”, is this a typo?
b. Page 1, Line 41: 3-9 hours is a range so it can be an average, unless this is the average values obtained from several studies? If so, the studies couldn’t have been homogenous and as such can’t be compared.
c. Page 1, Line 32-34: this sentence is confusing. Also, please note that a trough is different from a peak so I wouldn’t right “…trough (peak)…”!
d. Please replace “level” with “concentration” when referring to a drug concentration.
Regards and best wishes.
Author Response
Thank you for your precious comments.
Reviewer 2
1. The manuscript suffers from several editorial issues. I have described minor issues under Minor comments (below) but there appears to be paragraphs missing which makes an assessment of the scientific merit of this manuscript difficult (e.g. in the Methods section). I suggest enlisting the help of a proof-reader.
We are terribly sorry for the error in manuscript editing. The deleted part of Section 2.2. and contents for Section 2.3 were added in the revised manuscript (line 72~91). The whole contents of revised manuscript were carefully reviewed by all authors.
2. One of the most useful model evaluation techniques is a VPC plot. The GOF plots can be visually deceptive of model performance particularly with repeat data such as yours. I suggest you create a VPC plot
Thank you very much for your valuable comment. We fully agree with your opinion on the necessity of simulation-based diagnostics. Unfortunately, because the data for modeling includes various dosing regimens in each patient with different weight and kidney function, a general VPC with all dosing regimens containing all ranges of covariates was not possible. We, therefore, utilized the method of ‘pvcVPC’. The results were added in Methods (line 111~127) and Results (line 186~203) sections and a related reference (No. 13, Bergstrand M, Hooker AC, Wallin JE, and Karlsson MO. Prediction-corrected visual predictive checks for diagnosing nonlinear mixed-effects models. The AAPS J. (2011) 13:143-151.) was also added. In addition to VPC, 1,000 replication bootstrapping results were provided in Table 2 which was conducted as model assessment.
3. It is not clear, perhaps missing?, how R Shiny was used was a dosing tool once the PK model was developed and coded in R Shiny. It seems you obtained the EBEs from NONMEM then added them to R Shiny? If so, this isn’t sensible as the EBE pertain to the individual so you can only perform TDM on that individual only!?
4. Also not clear whether a Bayesian estimator was added to R Shiny and the nature of the estimator (was it MAP?).
5. Also related to above, which ODE solver was used? I think general technical information about the dosing tool are missing in this manuscript.
We thought it would be better to answer your question 3-5 in a comprehensive manner. Your comments were so helpful that we could reconstruct our manuscript to improve the readers’ understanding.
This open-source software using R Shiny was to develop for individual patient for VCM TDM. Building population PK model and covariate analysis were conducted by NONMEM. In the R shiny code, not only EBE equation and final estimates of both fixed-effect and random-effect parameters were built in, but also 2-compartment structural model and parameters with the objective function minimization methods were defined. In this process, we used R packages such as deSolve, trustOptim, DEoptim, plyr, and dplyr were used.
We believe that the user may obtain PK parameters of the patient of interest by entering clinical information and performing the minimization procedures in the value of objective function implemented in the R code. When patient information and dosing history is given with the observation time, the predicted concentration without the consideration of the ETA for the corresponding patient can be determined by the PK model. In addition, the system also obtains actual plasma concentrations as the user input values and reflect them as they are in the objective function. Thus, the only way to minimize the value of objective function is to modify ETA values (please refer Eq. (5) in the manuscript). This minimization process can be done with ‘DEoptim’ package of R.
We added details in the ‘2.6. R Shiny application for VCM TDM’ (line 136-157) how PK model built using NONMEM was implemented and applied to the R Shiny code.
6. There were a few typographical, grammar, and sentence structure issues throughout the manuscript. A couple of examples below. I recommend enlisting the help of a proof-reader.
a. Page 1, Line 36: “osteomyelitis meningitis”, is this a typo?
We are truly sorry for some errors in the manuscript, particularly in the ‘Introduction’ section. We revised sentences with unclear meaning and/or typographical problems (line 35-37, 40, 41, 47-48, and 50-51).
b. Page 1, Line 41: 3-9 hours is a range so it can be an average, unless this is the average values obtained from several studies? If so, the studies couldn’t have been homogenous and as such can’t be compared.
We revised the expression as you commented. Thank you for valuable input.
c. Page 1, Line 32-34: this sentence is confusing. Also, please note that a trough is different from a peak so I wouldn’t right “…trough (peak)…”!
We fully agree with your comment. The expression was removed.
d. Please replace “level” with “concentration” when referring to a drug concentration. We revised it.
Thank you for your comment which contributed to the scientific aspect of our manuscript. We checked the whole manuscript and revised it as you commented.
Reviewer 3 Report
Q: how the PK differs in Korean vs western population? I think the authors need to include this information to justify the analysis.
Q: in abstract, please add Q ( inter-compartmental clearance result) in line 25.
Q: in line 48, please define AUC24 and MIC on their first appearance.
Q: There is a "z" in line 72, what is this refereeing to?
Q: in line 72, the authors say that the mean of observation =0, how is that possible?
Q: in line 83: please define w2.
Q: no details are provided about the methodology to model BSV? how was it represented mathematically?
Q: line 93, please speficy if the bootstrap was parametric or non-parametric?
Q: line 95, please clarify what are the conditions you are referring to?
Q: line 102, please define eta for the audience on its first appearance for the BSV analysis.
Q: a VPC analysis needs to be included in the results as a qualification method.
Q: Figure 1, please use IWRES instead of |IWRES|, the use of absolute values hinders the ability to evaluate the distribution of points.
Q: again, in line 143, what are the conditions being referred to and why these conditions were chosen specifically?
Q: How many subjects are in group 4? and why the authors think the variability around group 4 ( in figure 2) increases with time?
Q: in the discussion, the authors need to show how their parameters estimate results and covariate analysis compare to the literature, and what would be the scientific reasons of these covariate analysis results.
Author Response
Thank you for your precious comments.
Reviewer 3
1. Q: how the PK differs in Korean vs western population? I think the authors need to include this information to justify the analysis.
Thank you for your comment which may enhance the readers’ understanding.
The major reason for additional PK study to improve TDM software is to optimize the performance to meet the needs in the clinical practice. As we mentioned in the manuscript line 52~54, previous reports showed that the current TDM software was not suitable for some patient groups. In addition, although it is very difficult to specify in the manuscript, there was a clinical suspicion on the inaccuracy of the current TDM software in Korean population suggested by the TDM team in the Seoul St.Mary’s hospital.
According to your comment, we rearranged the sentences (line 52-54) in the manuscript and added the difference between Western and Korean population (line 56). Basically, we were to confirm the PK parameter values in the Korean population who has different demographic characteristics in comparison to the Western population. In addition, with the ability to obtain data from patients who are undergoing HD or CRRT treatment, we were to identify the PK differences in those patient groups.
2. Q: in abstract, please add Q (inter-compartmental clearance result) in line 25.
The estimate for Q was added as you commented (line 23-25).
3. Q: in line 48, please define AUC24 and MIC on their first appearance.
Thank you very much for your comment. It was vital and we added the descriptions (line 47-48) Accordingly, other abbreviations are also explained (line 50-51).
4. Q: There is a "z" in line 72, what is this refereeing to?
We are terribly sorry for the error in manuscript editing. In the preparation stage, some contents for Section 2.2. and 2.3 were substituted with “z” (maybe, the ctrl + z was not correctly performed). The deleted part of Section 2.2. and contents for Section 2.3 were added in the revised manuscript (line 72~91). The whole contents of revised manuscript were carefully reviewed by all authors.
5. Q: in line 72, the authors say that the mean of observation =0, how is that possible?
This is the same problem mentioned in Q4 above. We revised it.
6. Q: in line 83: please define w2.
A number ‘2’ is a superscript and we corrected it.
7. Q: no details are provided about the methodology to model BSV? how was it represented mathematically?
This is the same problem mentioned in Q4 above. We revised it.
8. Q: line 93, please specify if the bootstrap was parametric or non-parametric?
Thank you for your precise comment. We indicated that it was non-parametric (line 129).
9. Q: line 95, please clarify what are the conditions you are referring to?
Thanks to your comment, we could add some details for the manuscript. We thought the specific condition was better to be added in the ‘Results’ section because the specific scenarios could not be determined in the study planning stages. Thus, we specified the actual simulation scenarios in the manuscript line 209-213.
10. Q: line 102, please define eta for the audience on its first appearance for the BSV analysis.
We used the term EBE as the major expression for ETA because the term is more appropriate. Also, we added descriptions for ETA for the readers (line 140-141)
11. Q: a VPC analysis needs to be included in the results as a qualification method.
Thank you very much for your valuable comment. We fully agree with your opinion on the necessity of simulation-based diagnostics. Unfortunately, because the data for modeling includes various dosing regimens in each patient with different weight and kidney function, a general VPC with all dosing regimens containing all ranges of covariates was not possible. We, therefore, utilized the method of ‘pvcVPC’. The results were added in Methods (line 111~127) and Results (line 186~203) sections and a related reference (No. 13, Bergstrand M, Hooker AC, Wallin JE, and Karlsson MO. Prediction-corrected visual predictive checks for diagnosing nonlinear mixed-effects models. The AAPS J. (2011) 13:143-151.) was also added. In addition to VPC, 1,000 replication bootstrapping results were provided in Table 2 which was conducted as model assessment.
12. Q: Figure 1, please use IWRES instead of |IWRES|, the use of absolute values hinders the ability to evaluate the distribution of points.
We fully agree with your comment. The figure was revised.
13. Q: again, in line 143, what are the conditions being referred to and why these conditions were chosen specifically?
This is the same problem mentioned in Q9 above. We revised it.
14. Q: How many subjects are in group 4? and why the authors think the variability around group 4 (in figure 2) increases with time?
As we mentioned above, the dataset consisted of highly variable population in the aspect of demographics and clinical status, we provided pvcVPC for the PK model. For Group 4 patients, the disease condition tends to be worse in comparison to the patient without HD therapy, so the medical treatment which may affect VCM PK (e.g. fluid therapy, time from the last HD treatment) may contribute the increased variability. However, the inflation in the prediction interval by time was a global problem of our model. This was because of insufficient number of observations in the extended treatment period (e.g. later than 500 hours from the initiation of VCM therapy). So, we added this as a limitation of our study in the discussion section (line 285-293).
15. Q: in the discussion, the authors need to show how their parameters estimate results and covariate analysis compare to the literature, and what would be the scientific reasons of these covariate analysis results.
Thank you for the comment. We added corresponding sentence in the discussion (line 249-251).
Round 2
Reviewer 2 Report
The authors have responded to questions, comments, and suggestions satisfactorily. I have no further comments.
Thank you and best regards.